# Entropy of Reissner–Nordström 3D Black Hole in Roegenian Economics

**DOI:** 10.3390/e21050509

**Published:** 2019-05-19

**Authors:** Constantin Udriste, Massimiliano Ferrara, Ionel Tevy, Dorel Zugravescu, Florin Munteanu

**Affiliations:** 1Faculty of Applied Sciences, Department of Mathematics-Informatics, University Politehnica of Bucharest, Splaiul Independentei 313, Bucharest 060042, Romania; tevy@mathem.pub.ro or; 2Di.Gi.ES, University Mediterranea of Reggio Calabria, Seconda Torre, Loc. Feo di Vito, 89125 Reggio Calabria, Italy; 3Institute of Geodynamics “Sabba S. Stefanescu”, Romanian Academy, Dr. Gerota 19-21, Bucharest 020032, Romania

**Keywords:** thermodynamics–economics dictionary, economic Einstein 4D PDEs, economic Schwarzschild-type metric, economic 3D black holes, economic entropy, 83C10, 83C57, 91B74, B41

## Abstract

The subject of this paper is to analyse the Mathematical Principia of Economic 3D Black Holes in Roegenian economics. In detail, we study two main problems: (i) mathematical origin of economic 3D black holes; and (ii) entropy and internal political stability depending on national income and the total investment, for economic Reissner–Nordström (RN) 3D black hole. To solve these problems, it was necessary to jump from macroeconomic side to microeconomic side (a substantial approach as they are so different), to complete the thermodynamics–economics dictionary with new entities, and to introduce the flow between two macroeconomic systems. The main contribution is about introducing and studying the Schwarzschild-type metric on an economic 4D system, together with Rindler coordinates, Einstein 4D partial differential equations (PDEs), and economic RN 3D black holes. In addition, we introduce some economic Ricci type flows or waves, for further research.

## 1. Introduction

The primary purpose of this article is to show that all economics principles, obtained via thermodynamics, are not in conflict but can be integrated seamlessly to complement each other. Although the results arise by combining and developing rapidly and successfully the previous notions, they do not depend on modeling/analysis tradeoffs. Our main contribution is a compositional mathematical language for Roegenian economics that combines thermodynamics, differential geometry and economics, along with a proof calculus and expressibility results. It is the basic objective of this paper to give answers to the following questions: What does a thermodynamics–economics morphism look like? What differential laws do we have in Roegenian economics? Do the gravitational models in physics have a correspondent in economics? What are the black holes in economics and are they similar to those in physics? In Roegenian economics, do the pseudo-Riemannian metric meanings make sense? Do Einstein’s PDEs make sense? What are the Schwarzschild metrics in Roegenian economics? What are the economic relationships between pair (economic entropy, stability of domestic policy) and pair (national income, total investment)? Does economic entropy have the basic properties of a general entropy? Can we estimate the entropy of the United States, China Entropy, etc.? Can economic flows and economic waves be characterised as PDE solutions? After a long period of gradual progress, Roegenian economics is now handled by Pfaff equations and PDE systems that produce the geometric equivalents. While the results are elegant and all background for the proofs is given, our proofs draw from many areas, including differential geometry, non-holonomic geometry, Einstein theory, solutions of partial differential equations, and mathematical analysis. All new points of view, which are the ones for results without citations, are included inline. Beyond the establishment of entropy of an economic black hole, the various sections of this paper implement in numerous ways the general theory borrowed from thermodynamics, physics and differential geometry.

In this paper, we develop the vision of Nicholas Georgescu Roegen, which links the economic phenomena of entropy. For this purpose, we identify a formal correspondence between economic processes and thermodynamic processes, with heuristic implications for the characterisation of the dynamics and evolution of economic systems. Then, on the basis of this formalism, we study the problem of establishing economic equilibrium conditions in the case of aggregation of initially stable and independent economic subsystems in a functional entity, e.g., the architecture of the European Community’s economy. Here, we expand the concept of the “black hole” from astrophysics to economics. This is possible by applying mathematical modeling to economic phenomena, the formalism commonly used to model entropic processes.

Our main result is contained in Theorem 3, Section 4: An economic black hole is characterised by entropy and internal political stability, depending explicitly on the (national) income and the total investment.

Our source of inspiration and the research tools cover the following topics.

Analytical continuation of Schwarzschild metric [1] underlines that some singularities in differential geometry of timespace are coordinate singularities and may be removed by an appropriate coordinate transformation.

Black holes in astrophysics [1,2,3,4,5,6,7,8] are regions of space in which the gravitational field is so strong that nothing can escape after falling beyond the horizon event.

Complexity [9] refers to theoretical and methodological foundations of the science of complex systems.

Controllability of a non-holonomic macroeconomic system [10,11,12] shows that Roegen structured economic systems respond best to optimal commands.

Econophysics [13,14,15,16,17,18,19] applies methods and principles of physics to economics. This theory includes the role of temperature in economic exchange, a partial history of attempts to connect physics and economics, and a conceptual thermodynamics–economics discussion. However, it is fully recognised that a general theory of economics is of little direct use in treating concrete problems of reality. In economics, pointwise forecasting is impossible because this requires the use of mathematical functions of point. The name “econophysics” was coined by E. Stanley, a physicist from Boston University, in 1996.

Gravity models of trade [20,21,22,23] mimic the gravitational patterns in physics and predict flows between two macroeconomic systems.

Complexplot3d applies to functions $\left(\cdot ,\cdot \right):R^{2}\to R^{2}$, namely, complexplot3d plots the first component while colouring the graphic using the second component. This idea has a profound meaning in Roegenian economics.

Now, let us explain our tools that led us to the results of this paper.

A black hole in economics has two facets: (i) as a metaphor in economics, it is a business activity or product on which large amounts of money are spent, but that does not produce any income or other useful result [24]; and (ii) in rising poverty, it is a small part of a global economic system where the total income created is so strong that nothing can escape after falling beyond the horizon event, i.e., an economics image from astrophysics via a morphism [8,25,26,27].

Economic cycles of Carnot type [28] are introduced into Roegenian economics, using a thermodynamics–economics dictionary, where (G,I,E,P,Q) are economic variables (*G* is growth potential, *I* is internal politics stability, *E* is entropy, *P* is price level (inflation), and *Q* is volume, structure, or quality) and the Gibbs–Pfaff fundamental equation dG−IdE+PdQ=0 on $R^{5}$ describes an economic distribution. In this paper, empirical elements emerge with economic support.

Non-holonomic approach of economic systems [11,25,29,30,31] means economics via Pfaff equations, together with techniques from differential geometry.

Phase diagram for Roegenian economics [32] underlines a triple point of an economic system as a pair (internal political stability, price level) at which the three phases (inflation, monetary policy of liquidity, and income) of that economic system coexist in economic equilibrium. The diagram of economic phases reveals the meaning of the states of an economic system evolving.

Roegenian economics [26,27,28,30,33] creates new mathematical models used in economic theorising, but also with the delicate epistemological problem of economics, starting from thermodynamics–economics morphism. The intention is to highlight the main elements of the Roegenian economics by presenting it in contrast with the classical economics described by the neoclassical paradigm. The idea has the root in a book [34] by Nicholas Georgescu-Roegen who argued that all natural resources are irreversibly degraded when put to use in economic activity. From natural products with little entropy, mankind has created goods or waste with great entropy. The imbalance diminishes only through political and economic decisions. The central problem is that economic scarcity is the reflection of the Entropy Law, which is the most economic in nature of all natural laws. Roegen introduced into the economy the concept of entropy, similar to that in thermodynamics, creating the foundation that later developed in the evolutionary economy. His research has also contributed significantly to the development of bio-economy and eco-economy.

## 2. Thermodynamics–Economics Dictionary

In the following, we reproduce the correspondence between the characteristic state variables and the laws of thermodynamics with the macroeconomics as described in our papers. They also allow us to study in economics the idea of an “economic black hole” with a similar meaning to the one in astrophysics [2,3,4,5]. This idea is developed in this paper as the mathematical origin of economic 3D black holes and as the entropy of Reissner–Nordström 3D black holes in Roegenian economics.

Therefore, each economic system should be similar to a thermodynamic system, in the sense that thermodynamics potentials and laws have correspondence in economics.

In our sense, the dictionary means a table that lists the words of a thermodynamic language and their correspondents in economics, associating “elements” that behave similarly. It is similar to a morphism (a structure-preserving map from one mathematical structure to another one of the same type). The information about usage of the dictionary is given in the text of the paper.

THERMODYNAMICS
ECONOMICSU = internal energy…G = growth potentialT = temperature…I = internal political stabilityS = entropy…E = entropyP = pressure…P = price level (inflation)V = volume…Q = volume, structure, qualityM = total energy (mass)…Y = national income (income)Q = electric charge…I = total investmentJ = angular momentum…J = economic angular momentum       (spin)
       (economic spin)M = M(S,Q,J)…Y = Y(E,I,J)$\mu_{k}$ = chemical potential…$\nu_{k}$ = economical potential$N_{k}$ = number of moles…$N_{k}$ = number of economic molesW = mechanical work…W = wealth of the systemQ = heat…q = stock market$T_{H}=\frac{\partial M}{\partial S}$ = Hawking temperature…$I_{BH}$ = BH-internal political stability*G* = Newton constant…G = universal economic constant*c* = light velocity…*c* = maximum universal exchange speed*ℏ* = normalised Planck constant…*ℏ* = normalised economic quantum

The last three lines in the previous dictionary are introduced here for the first time (and are required by economic Schwarzschild metric and economic 3D black hole theory).

### 2.1. Thermodynamics Differential Laws

The process variables W=
*mechanical work* and *Q* = *heat* are introduced into Carathéodory thermodynamics by dW=PdV (the *first law*) and by elementary heat, respectively, dQ=TdS (differential equality), for reversible processes, or dQ<TdS (differential inequality), for irreversible processes (*second law*).

*The Gibbs–Pfaff fundamental equation in thermodynamics* is $dU-TdS+PdV+\sum_{k}\mu_{k}dN_{k}=0$ (a combination of the *first law* and the *second law* in thermodynamics).

The *third law* of thermodynamics states that $\lim_{T\to 0}S=0$.

### 2.2. Roegenian Economics Differential Laws

The process variables in economics *W* = *wealth of the system*, *q* = *stock market* are defined by dW=Pdq (*first economic law, elementary wealth in the economy*) and dq=IdE or dq<IdE (*second economic law, elementary production of commodities*). A *commodity* is an economic good, a product of human labor, with a utility in the sense of life, for sale/purchase on the market in the economy.

Let us accept that the *Gibbs–Pfaff fundamental equation of economy* is $dG-IdE+PdQ+\sum_{k}\nu_{k}dN_{k}=0$ (combination of the *first law* and the *second law* in economics).

The *third law of economics*
$\lim_{I\to 0}E=0$ says that “if the internal political stability *I* tends to 0, the system is blocked, meaning entropy becomes E=0, equivalent to maintaining the functionality of the economic system must cause disruption”.

The long term association between economics and thermodynamics can be strengthened with new tools based on extensions of the foregoing morphism. This morphism allows the transfer of information from thermodynamics to economics, keeping the background of each discipline, which we believe was suggested by Roegen [34]. Of course, this new idea of thermodynamics–economics morphism will produce new concepts in econophysics (see [27,32]).

A macroeconomic system based on a Gibbs–Pfaff equation is controllable (see [11]).

**Definition** **1.**
*An economy which is structured similarly to thermodynamics, via the previous dictionary, is called Roegenian economics.*


### 2.3. Gravity Models in Physics and Economics

Newton’s law of universal gravitation states that every particle attracts every other particle in the universe with a force which is directly proportional to the product of their masses and inversely proportional to the square of the distance between their centres. Thus, the equation for the universal gravitation takes the form
$$F=G\;\;\frac{m_{1}m_{2}}{r^{2}},$$
where *F* is the gravitational force acting between two objects, $m_{1}$ and $m_{2}$ are the masses of the objects, *r* is the distance between the mass centres of the objects, and *G* is the gravitational constant.

The normalised Planck constant *ℏ* is a physical constant that is the quantum of action (central notion in quantum mechanics). The Planck constant represents the proportionality between the momentum and the quantum wavelength of not only of the photon, but the quantum wavelength of any particle.

The previous dictionary adds to “gravity models of trade” [20,21,22,23] another more significative flow between two macroeconomic systems.

**Definition** **2.**
*The economic centre of gravity of a specific economic system is the region with the largest contribution to (national) income.*


In this sense, the distance *r* between two economic systems means the economic distance between two centres of economic gravity.

The flow between (national) incomes shows that every macroeconomic system attracts every other macroeconomic system by a force acting between the economic centres of gravity. The strength of this force is proportional to the product of the two national incomes, and inversely proportional to the square of the distance between them.

**Definition** **3.**
*Suppose G stands for the universal economic constant (an economic constant of proportionality), Y stands for the (national) income, and r stands for the economic distance between the two macroeconomic systems that are studied. The formula*
$$F=G\;\frac{Y_{1}Y_{2}}{r^{2}},$$
*defines the flow between (national) incomes.*


The normalised economic quantum *ℏ* is the economic quantum of action.

**Remark** **1.**
*In our idea, the gravity constant and the Planck constant are related to the quantisation of the dynamic quantities related to the microscopic world. This aspect finds a scientific motivation also in the social sciences such as economics: in mechanics and quantum physics, the same energies that act by producing effects and externalities in the real world at the same time play a role in socioeconomic mechanism designs in real life. The parallelism between light speed and a concrete quantity as universal exchange speed, is rational and absolutely well-founded. This exchange-type could model the digital economic transitions among agents by the Internet. In our model, we have fixed this constant to 1 (geometrised units), with the main aim to discuss the scientific role played by this parameter, which in this case produces no effort.*


## 3. Mathematical Origin of Economic 3D Black Holes

We recall what economists commonly call “economic black holes” [24]: (i) a Keynesian black hole (in terms of liquidity trap) means a business activity or product on which large amounts of money are spent, but that does not produce any income or other useful result; and (ii) a liquidity black hole means that trading financial assets becomes prohibitively expensive and the asset price collapses (e.g., 1997 Asian crisis, 1998 Russian debt/LTCM crisis, and 2007 subprime mortgage crisis).

In our papers [8,25,26,32], we described a new concept of “economic 3D black hole” as a small part of a global economic system where the total income created is so strong that nothing can escape after falling beyond the horizon event (rising poverty). This kind of economic black hole is depicted with entropy *E*, national income (income) *Y*, total investment I and economic spin *J*. The national income is so large that it attracts all the economic resources of its neighbours. This is in fact the image in the previous dictionary of a thermodynamics black hole.

Looking again on some papers regarding thermodynamics 3D black holes (especially [2]) and on our papers regarding economic 3D black holes, we now try to explain the infinitesimal roots of economic 3D black holes, i.e., the mathematical principia of economic 3D black holes. To do that, we use an arithmetic economic time-space $R\times R^{3}$, with four dimensions. A point in this space has the coordinates (t,G,I,E), where *t* is the time, and (G,I,E) are economic variables (*G* is potential growth, *I* is internal political stability, and *E* is entropy), stripped of their true meaning and units of measure. Geometrically, the shorthand for all selected coordinates is $x^{\mu }$, where the index μ take values 0;1;2;3. National income *Y*, total investment I, and economic spin *J* are economic parameters.

To understand the relevant parameters and the geometry of economic black holes, we add some economic–geometric ingredients: the universal economic constant G, an economic metric $g_{\mu \nu }$ of signature (−;+;+;+) (which captures all the geometric and causal structures of economic time-space), the determinant of the metric $g=\det(g_{\mu \nu })$ (which is a negative number), the arc-length square $ds^{2}=g_{\mu \nu }dx^{\mu }dx^{\nu }$, the Ricci tensor field $R_{\mu \nu }$, and the Ricci scalar $R=g^{\mu \nu }R_{\mu \nu }$. In addition, we set c=1 (see geometrised units). The pseudo-Riemannian approach to economics is not only a mathematical curiosity but also a useful technique to solve actual problems, since it might deliver local information of economic systems relying solely on global data.

Similar to the Einstein–Maxwell theory in thermodynamics, we introduce the economic action

$$S=\frac{1}{16\pi G}\;\;\int R\sqrt{-g}\;\;d^{4}x.$$

**Theorem** **1.**
*The Euler–Lagrange PDEs are the vacuum 4D Einstein PDEs*
$$R_{\mu \nu }=\frac{R}{2}\;\;g_{\mu \nu }.$$


**Proof.** The technique of obtaining the Euler–Lagrange PDEs is well-known. Equating to zero the variation, i.e.,
$$0=\delta S=\frac{1}{16\pi G}\;\;\int \left(\frac{\delta R}{\delta g^{\mu \nu }}+\frac{R}{\sqrt{-g}}\frac{\delta \sqrt{-g}}{\delta g^{\mu \nu }}\right)\delta g^{\mu \nu }\;\;\sqrt{-g}\;\;d^{4}x,$$
taking $\delta g^{\mu \nu }$ arbitrary, we obtain the Euler–Lagrange PDEs (the equation of motion for the metric field),
$$\frac{\delta R}{\delta g^{\mu \nu }}+\frac{R}{\sqrt{-g}}\frac{\delta \sqrt{-g}}{\delta g^{\mu \nu }}=0$$
or the vacuum Einstein PDEs
$$R_{\mu \nu }=\frac{R}{2}\;\;g_{\mu \nu },$$
with the unknowns $g_{\mu \nu }$ (components of the metric). □

### 3.1. Schwarzschild Type Metric on the 4D Economic System

The Euler–Lagrange PDEs $\frac{\partial L}{\partial g^{\mu \nu }}=0$, attached to the economic Lagrangian
$$L=\frac{1}{16\pi G}\;\;R\sqrt{-g},$$
are vacuum Einstein PDEs with the unknowns $g_{\mu \nu }$ (components of the pseudo-Riemannian metric). Consider the economic Schwarzschild metric $g_{\mu \nu }$, which is a spherically symmetric, static solution of previous Einstein PDEs. This metric is expected to describe the geometry of economic time-space outside or inside an economic black hole.

Let us use G as the universal economic constant, *Y* as the national income (parameter), *t* as the time, *r* as the radial coordinate, and Ω as the solid angle on a two-sphere. Then, economic Schwarzschild metric solution $g_{\mu \nu }$ is given by the arc-length square
$$ds^{2}=-\left(1-\frac{2GY}{r}\right)dt^{2}+{\left(1-\frac{2GY}{r}\right)}^{-1}dr^{2}+r^{2}\;d\Omega^{2}.$$

If r>2GY (exterior), then the signature of the metric $g_{\mu \nu }$ is (−,+,+,+). If 0<r<2GY (interior), then the signature of this metric $g_{\mu \nu }$ is (+,−,+,+). The singularity r=0 is known as a curvature singularity and is irremovable. The metric appears also to be singular at r=2GY because $g_{00}=0$ (vanish) and $|g_{rr}|\to \infty$ (diverge). Let us show that the singularity r=2GY is a coordinate singularity and may be removed by an appropriate coordinate transformation. For example, the coordinate transformation
$$\left(t,r,\theta ,\varphi \right)\to \left(u,r,\theta ,\varphi \right),\;\;u=t-r-2GY\;\;\ln\left(r-2GY\right),\;\;du=-dt-{\left(1-\frac{2GY}{r}\right)}^{-1}dr$$
produces
$$ds^{2}=-\left(1-\frac{2GY}{r}\right)du^{2}-2dudr+r^{2}\left(d\theta^{2}+sin^{2}\theta d\varphi^{2}\right).$$

In the new coordinates (u,r,θ,φ), the components of the metric are non-singular at r=2GY. Moreover, the previous exterior Schwarzschild solution may be analytically continued across the surface given by the equation r=2GY [1].

To better understand the nature of this apparent singularity, let us examine the geometry closer to the *event horizon spherical surface*
r=2GY of the exterior Schwarzschild solution. Much of the interesting economics (physics) having to do with the quantum properties of economic black holes comes from the region near the event horizon.

**Theorem** **2.**
*The pseudo-Riemannian metric*
$$ds^{2}=-\left(1-\frac{2GY}{r}\right)dt^{2}+{\left(1-\frac{2GY}{r}\right)}^{-1}dr^{2}+r^{2}\;d\Omega^{2}.$$
*is approximated by the pseudo-Riemannian metric*
$$ds^{2}=-\frac{\rho^{2}}{16G^{2}Y^{2}}\;\;dt^{2}+d\rho^{2}+{\left(2GY\right)}^{2}d\Omega^{2}.$$


**Proof.** To focus on the near horizon geometry in the region r−2GY≪2GY, let us define r−2GY=ξ, so that, when r→2GY, we have ξ→0. Then, the metric takes the form
$$ds^{2}=-\frac{\xi }{2GY+\xi }\;dt^{2}+\frac{2GY+\xi }{\xi }\;d\xi^{2}+{\left(2GY+\xi \right)}^{2}\;d\Omega^{2}.$$Taking into account the inequality $\frac{\xi }{2GY}\ll 1$, we find the approximation
$$ds^{2}=-\frac{\xi }{2GY}\;dt^{2}+\frac{2GY}{\xi }\;d\xi^{2}+{\left(2GY\right)}^{2}\;d\Omega^{2},$$
up to corrections that are of order $\frac{1}{2GY}$. Introducing a new coordinate ρ, by $\rho^{2}=\left(8GY\right)\xi$, so that $\frac{2GY}{\xi }d\xi^{2}=d\rho^{2}$, the metric takes a new form
$$ds^{2}=-\frac{\rho^{2}}{16G^{2}Y^{2}}dt^{2}+d\rho^{2}+{\left(2GY\right)}^{2}d\Omega^{2}.$$From this form of the metric, it is clear that the coordinate ρ measures the geodesic radial distance. Note that the geometry factorises. One factor is a two-sphere of radius 2GY and the other is the (ρ,t) space
$$ds_{2}^{2}=-\frac{\rho^{2}}{16G^{2}Y^{2}}dt^{2}+d\rho^{2}.$$
 □


In the next subsection, we show that this 1+1-dimensional time-space is just a Minkowski space written in funny coordinates called the *Rindler coordinates*.

### 3.2. Rindler Coordinates

To understand Rindler coordinates and their relation to the near horizon geometry of the economic black hole, let us start with the 1+1 Minkowski space with the usual flat Minkowski metric $ds^{2}=-dT^{2}+dX^{2}.$ Introducing the light-cone coordinate, U=T+X, V=T−X, this metric becomes $ds^{2}=-dUdV$. Now, we pass to Rindler coordinates (u,v), via an exponential change $U=\frac{1}{\kappa }\;e^{\kappa u}$, $V=-\frac{1}{\kappa }\;e^{-\kappa v}$, in which $ds^{2}=-e^{\kappa (u-v)}\;dudv$.

We change again the coordinates via $u=t+x,v=t-x,\rho =\frac{1}{\kappa }\;e^{\kappa x}.$ Then, the metric becomes $ds^{2}=-\rho^{2}\kappa^{2}\;dt^{2}+d\rho^{2}.$ Comparing with $ds_{2}^{2}$, we obtain the *surface economic gravity*$\kappa =\frac{1}{4GY}$ of the economic black hole.

For the economic Schwarzschild solution, one can think of it heuristically as the economic Newtonian acceleration $\frac{GY}{r_{H}^{2}}$ at the *horizon radius*
$r_{H}=2GY$. The surface economic gravity κ and the horizon radius $r_{H}$ play an important role in describing the sense of an economic black hole. This analysis proves that the economic Schwarzschild time-space near the surface r=2GY is not singular at all. After all, it looks exactly like a Cartesian product between a flat Minkowski space and a sphere of radius 2GY. Thus, the curvatures are inverse powers of the radius of curvature 2GY and hence are small for large 2GY.

### 3.3. Economic Schwarzschild Radius

An economic singularity or time-space singularity is a location in time-space where the economic metric field of an economic system becomes infinite in a way that does not depend on the coordinate system.

The quantities used to measure the economic field strength are the scalar invariant curvatures of the time-space, which includes a measure of the economic density $\frac{\partial Y}{\partial Q}$.

Let us give some economic information borrowed from Schwarzschild metric theory. The economic Schwarzschild radius is given now as $r_{s}=\frac{2GY}{c^{2}}$, where G is the universal economic constant, *Y* is the national income and *c* is the maximum universal exchange speed. The economic Schwarzschild radius is an economic parameter that appears in the economic Schwarzschild solution to Einstein’s field equations, corresponding to the radius defining the economic event horizon of an economic Schwarzschild black hole. In fact, it is a characteristic radius associated with every (national) income.

**Remark** **2.**
*Let $R_{\alpha \beta \gamma \delta }$ be the curvature tensor field. An important quantity is the economic invariant given by*
$$R^{\alpha \beta \gamma \delta }R_{\alpha \beta \gamma \delta }=\frac{12r_{s}^{2}}{r^{6}}=\frac{48G^{2}Y^{2}}{c^{4}r^{6}},$$
*where r is the radial coordinate (measured as the circumference, divided by 2π, of a great circle of the sphere centred around the economic system), and $r_{s}$ is the economic Schwarzschild radius of the economic system, a scale factor which is related to its (national) income Y by $r_{s}=\frac{2GY}{c^{2}}$, where G is the universal economic constant.*


## 4. Economic Reissner–Nordström (RN) 3D Black Hole

For this section, we need an economic action containing an antisymmetric tensor field $F_{\mu \nu }$ (similar to the electro-magnetic field strength). Explicitly, we set c=1 and we use the following notations: G is universal economic constant, $R_{\mu \nu }$ is the Ricci tensor field, $R=g^{\mu \nu }R_{\mu \nu }$ is the Ricci scalar of the metric $g_{\mu \nu }$, the negative number $g=\det(g_{\mu \nu })$ is the determinant of the metric $g_{\mu \nu }$, and $F_{\mu \nu }$ is an economic field strength with $F^{2}=g^{\mu \lambda }g^{\nu \sigma }F_{\mu \nu }F_{\lambda \sigma }$.

We introduce the economic action (multiple integral functional)
$$\frac{1}{16\pi G}\;\;\int R\sqrt{-g}\;\;d^{4}x-\frac{1}{16\pi }\;\;\int F^{2}\sqrt{-g}\;\;d^{4}x$$
associated to the Lagrangian
$$L=\frac{1}{16\pi G}\;\;R\sqrt{-g}-\frac{1}{16\pi }\;\;F^{2}\sqrt{-g}.$$

Let I be the total investment, *Y* the national income, and $F_{tr}=\frac{I^{2}}{r^{2}}$ the non-zero economic field strength component. The pseudo-Riemannian metric of the economic system is now given via arc-length square
$$ds^{2}=-\left(1-\frac{2Y}{r}+\frac{I^{2}}{r^{2}}\right)dt^{2}+{\left(1-\frac{2Y}{r}+\frac{I^{2}}{r^{2}}\right)}^{-1}dr^{2}+r^{2}d\Omega^{2}.$$

We can recover the formulae for economic Schwarzschild metric taking the limit I→0.

The most general static, spherically symmetric, charged solution of the Einstein–Maxwell theory gives the economic Reissner–Nordström (RN) 3D black hole.

From the previous metric, we see that the event horizon for this solution is located where $g_{tt}=0$, or $1-\frac{2Y}{r}+\frac{I^{2}}{r^{2}}=0$, or $r_{\pm }=Y\pm \sqrt{Y^{2}-I^{2}}$. Thus, $r_{+}$ defines the outer horizon of the economic black hole and $r_{-}$ defines the inner horizon of the economic black hole. The area of the event horizon is given by $4\pi r_{+}^{2}$ (sphere). For an economic Schwarzschild black hole, the area is $A=16\pi G^{2}Y^{2}$, and the economic surface gravity is $\kappa =\frac{1}{4GY}$, where G is the universal economic constant.

Any economic system has an “internal political stability” and hence it has “economic entropy”.

In 1974, the British physicist Stephen Hawking discovered that black holes have a characteristic temperature and are therefore capable of emitting radiation. Let us show that something similar happens to the economic black holes, i.e., there exists a characteristic “BH-internal political stability”, namely, a marginal inclination to entropy $\frac{\partial Y}{\partial E}$.

Let *ℏ* be the economic Planck constant. The “BH-internal political stability” and the “economic entropy” are given in terms of the surface economic gravity and horizon area by the formulae
$$I_{BH}=\frac{\kappa \hbar }{2\pi },\;\;E=\frac{Ac^{3}}{4G\hbar },$$
where *A* means area of the economic black hole. Using geometrised units where G=1,ℏ=1,c=1, we can formulate the following.

**Theorem** **3.**
*An economic black hole has two characteristics depending on the (national) income and the total investment:*

*(i) The entropy*
$$E=\pi r_{+}^{2}=\pi {\left(Y+\sqrt{Y^{2}-I^{2}}\right)}^{2}.$$

*(ii) The BH-internal political stability*
$$I_{BH}=\frac{\kappa }{2\pi }=\frac{\sqrt{Y^{2}-I^{2}}}{2\pi \left[2Y\left(Y+\sqrt{Y^{2}-I^{2}}\right)-I^{2}\right]}.$$


**Hints**: (i) Identify the horizon for the previous economic metric and examine the near horizon geometry to show that it has the two-dimensional Rindler space-time as a factor.

(ii) Using the relation with the Rindler geometry, we determine the economic surface gravity κ for the economic Schwarzschild black hole and thereby determine the internal political stability of the economic black hole.

(iii) In the extremal limit Y→I, the internal political stability vanishes but the entropy has a nonzero limit.

(iv) Finally, for the extremal Reissner–Nordström economic black hole, the near horizon geometry is of the form $AdS_{2}\times S^{2}$, i.e., (r,t) is a two-dimensional anti-de Sitter ($AdS_{2}$) and the second factor is the two-sphere $S^{2}$.

**Corollary** **1.**
*(i) The entropy of the economic black hole is a convex function on the space {Y,I},Y≥I≥0.*

*(ii) The spaces {E,I},E≥I≥0, and {Y,I},Y≥I≥0, are diffeomorphic equivalent.*


The total entropy (total BH-internal political stability) of the economic black hole is obtained by integration.

### 4.1. Empirical Analysis on Roegenian Economics

**Entropy approximation** Since the total investment is a subunit per cent of GDP, we accept $\frac{I}{Y}<<1$. Then, ${\left(1-\frac{I^{2}}{Y^{2}}\right)}^{1/2}$ is approximated by $1-\frac{1}{2}\frac{I^{2}}{Y^{2}}$, again by 1, and hence the economic entropy is estimated by $E≅4\pi Y^{2}$. Based on UN and IMF estimations, the United States has the largest GDP in the world at US$ 18,624,475 trillion (UN) and US$ 21,410,230 trillion (IMF). The second largest GDP is China’s at $ 11,218,281 (UN) and $ 15,543,710 (IMF). Automatically, we have estimations for entropy of particular economic Reissner–Nordström 3D Black Holes, as for example $E_{US}$ and $E_{C}$.

**Maple subroutine “complexplot3d”** Let us consider either the function $\left(E,I\right):D\subset R^{2}\to R^{2}$ or the function $\left(I,E\right):D\subset R^{2}\to R^{2}$. The plot of the first component coloured by the second component highlights the entropic areas of economic interest (Figure 1), and the areas of internal political stability influenced by entropy (Figure 2).

### 4.2. Economic 4D Einstein PDEs and the Stress–Energy Tensor Field

Consider c=1 and denote 2κ=16πG. Suppose that we use a full economic action
$$S=\int \left(\frac{1}{2\kappa }\;R+L\right)\;\sqrt{-g}\;\;d^{4}x,$$
where L describes any economic fields appearing in the economics theory.

**Theorem** **4.**
*The Euler–Lagrange PDEs are the vacuum Einstein PDEs*
$$R_{\mu \nu }-\frac{R}{4}\;\;g_{\mu \nu }=kT_{\mu \nu }.$$


**Proof.** Again, we use a well-known technique. Then, the action principle tells us that the variation of this action with respect to the inverse metric is zero, yielding
$$0=\delta S=\int \left[\frac{1}{2\kappa }\left(\frac{\delta R}{\delta g^{\mu \nu }}+\frac{R}{\sqrt{-g}}\frac{\delta \sqrt{-g}}{\delta g^{\mu \nu }}\right)+\frac{1}{\sqrt{-g}}\frac{\delta (\sqrt{-g}L)}{\delta g^{\mu \nu }}\right]\delta g^{\mu \nu }\;\sqrt{-g}\;\;d^{4}x.$$Since this equality should hold for any variation $\delta g^{\mu \nu }$, we find
$$\frac{\delta R}{\delta g^{\mu \nu }}+\frac{R}{\sqrt{-g}}\frac{\delta \sqrt{-g}}{\delta g^{\mu \nu }}=-2\kappa \;\;\frac{1}{\sqrt{-g}}\frac{\delta (\sqrt{-g}L)}{\delta g^{\mu \nu }}.$$Since $\frac{\delta R}{\delta g^{\mu \nu }}=R_{\mu \nu }$, these are the equations of motion for the metric tensor field $g_{\mu \nu }$, i.e., Einstein PDEs. The right hand side is proportional to the economic stress–energy tensor field
$$T_{\mu \nu }=\frac{-2}{\sqrt{-g}}\frac{\delta (\sqrt{-g}L)}{\delta g^{\mu \nu }}=-2\;\;\frac{\delta L}{\delta g^{\mu \nu }}+g_{\mu \nu }L.$$
 □


## 5. Economic 4D Einstein PDEs and Ricci Type Flows or Waves

Let $g_{\mu \nu }\left(x\right)$ be the components of a general economic metric and $R_{\mu \nu }$ the associated Ricci tensor on the time-space $R\times R^{3}$. The first component of $x=(x^{0},x^{1},x^{2},x^{3})$ is the time $x^{0}=t$. The solutions $g_{\mu \nu }\left(x\right)$ of the Einstein PDEs
$$R_{\mu \nu }=\frac{R}{2}\;\;g_{\mu \nu }$$
are “waves” with respect to $x^{0}=t$.

Suppose that the components of the economic metric $g_{\mu \nu }\left(x\right)$ do not depend explicitly on *t*. Then, we introduce a Ricci type flow g(x,τ) satisfying $\frac{\partial g_{\mu \nu }}{\partial \tau }=-2R_{\mu \nu }$ and a Ricci type wave g(x,τ) by $\frac{\partial^{2}g_{\mu \nu }}{\partial \tau^{2}}=-2R_{\mu \nu }.$ An evolution metric g(x,τ), starting from $g_{\mu \nu }\left(x\right)$, determines an evolution of all geometric ingredients (connection, geodesics, curvature tensor field, Ricci tensor field, and scalar curvature).

On the other hand, we can introduce the economic flow $\frac{\partial g_{\mu \nu }}{\partial Y}=-2R_{\mu \nu }$ and the economic wave $\frac{\partial^{2}g_{\mu \nu }}{\partial Y^{2}}=-2R_{\mu \nu },$ where *Y* is the (national) income. The last two PDEs are ingredients for producing economic metrics with special properties.

The Ricci flow and the Ricci wave were introduced as tools to address a variety of non-linear problems in differential geometry and, in particular, the uniformisation of compact Riemannian manifolds. The deformation variable *t* (or τ) that otherwise appears ad hoc in mathematics, have meaning in physics, but produces difference of philosophy among economics, physics and mathematics regarding the applicability of the Ricci flow or the Ricci wave. In further papers, we shall be looking for economic interpretations of all previous flows and waves.

Open problem: The pseudo-Riemannian metric flows and pseudo-Riemannian metric dynamics (wave) generated by the covariant derivative of geometric vector fields (Killing vector field, conformal vector field, irrotational vector field, solenoidal vector field, and harmonic vector field) on pseudo-Riemannian manifolds need to be studied. Do such geometrical theories have any economic meaning?

## 6. Discussion

Our papers are significant contributions to Roegenian economics because they set a vision and provide a framework for economics similar to thermodynamics based on a dictionary. In time, we developed and studied the following ideas: extrema with non-holonomic constraints [29], non-holonomic economic systems [30], economic geometric dynamics [25], black hole geometric thermodynamics [6,7], thermodynamics versus economics [33], multi-time optimal economic growth [10,35], black hole models in economics [8], geobiodynamics and Roegen type economy [26], non-holonomic geometry of economic systems [31], controllability of a non-holonomic macroeconomic system [11], optimal control on non-holonomic black holes [12], phase diagram for Roegenian economics [32], geobiodynamics and Roegenian economic systems [27], and economic cycles of Carnot type [28]. We interpret this collection of papers as a call to the economics–physics–mathematics community to respond to the current political forces that (inappropriately) shape our life. This article examines again the arguments and concludes that a proper dictionary may provide a productive way forward in econophysics. Of course, economics based on thermodynamics does not invalidate the traditional economy but confirms new things that until now could not be explained directly.

Our theory addresses the role of the mathematical context via proper dictionary, a topic absent from most mathematical papers. An implication of this argument is the need to strengthen the quality of the mathematics component in economics. The mathematical concepts turn up in entirely unexpected connections. Moreover, they often permit an unexpectedly close and accurate description of the economic phenomena in these connections. In addition, because we do not always understand the reasons of Roegenian economics usefulness, we appreciate that a theory formulated in terms of mathematical concepts is uniquely appropriate.

Mathematics and physics play important roles in economics, but it is not so easy to recognise this. The uniqueness of the theories of mathematics and physics must impose the same thought in economics. A proper answer to Roegenian economics would require elaborating theoretical work which has not been totally undertaken up to date. Our paper has a high degree of complexity because it uses techniques and ideas from differential geometry and thermodynamics to produce non-contradictory information in economics.

In this context, we show that an economic black hole has two characteristics depending on the (national) income and the total investment: (i) the entropy; and (ii) the BH-internal political stability. The formulas found by us generate a diffeomorphism between the economic spaces {E,I},E≥I≥0, and {Y,I},Y≥I≥0, i.e., the pairs (entropy, internal political stability) and (national income, total investment) are deeply economically interconnected. The Maple subroutine “complexplot3d” applied to the pair (E,I) plots the first component *E* coloured by the second component *I*, highlighting entropic areas of economic interest, while applied to the pair (I,E) gives areas of internal political stability influenced by entropy. Future research will clarify the economic sense of these statements.

## Figures and Tables

**Figure 1 entropy-21-00509-f001:**
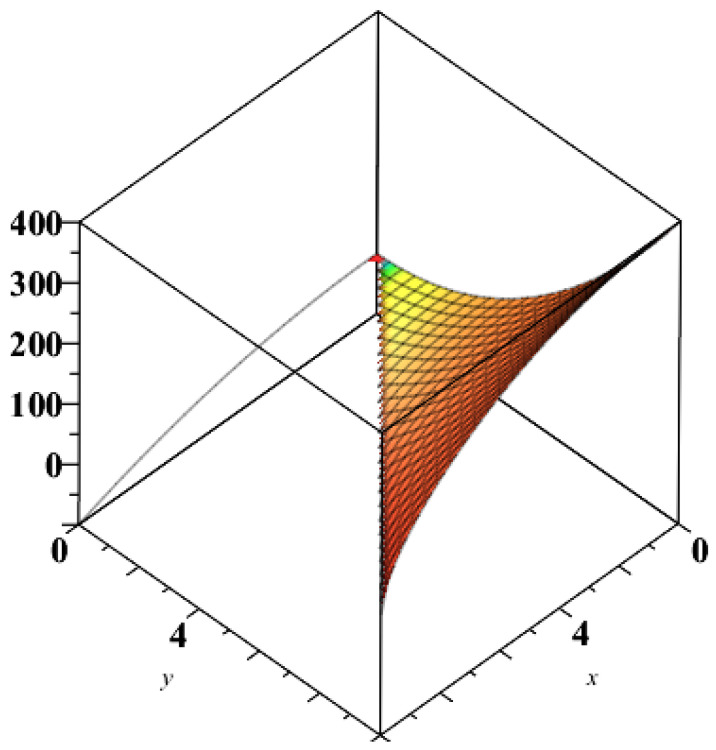
Plot entropy coloured by internal political stability.

**Figure 2 entropy-21-00509-f002:**
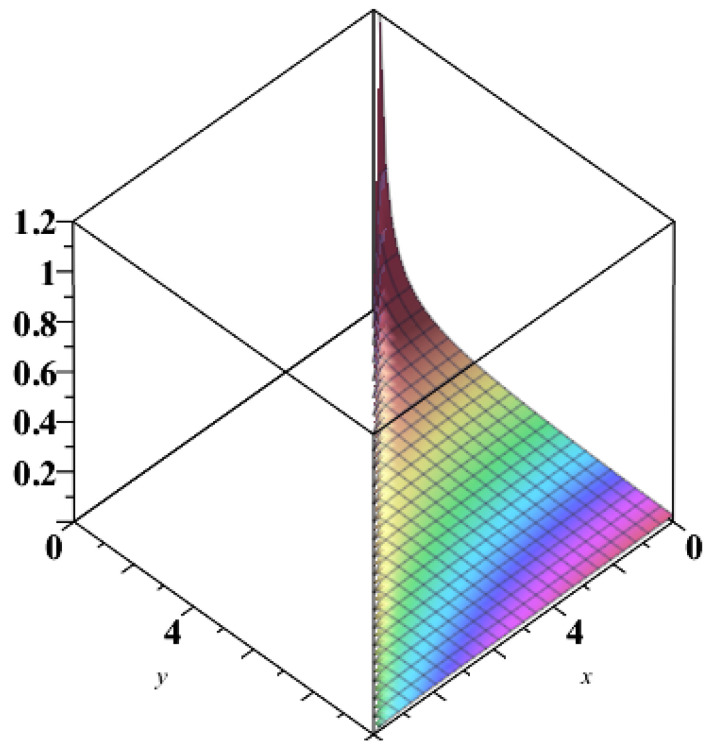
Plot internal political stability coloured by entropy.

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
