# Peer review of "Entropy of Reissner–Nordström 3D Black Hole in Roegenian Economics"

_entropy, 2019, doi:10.3390/e21050509_

Round 1

Reviewer 1 Report

This manuscript is interesting, and I have reviewed the original version  a few days ago. The authors have made some improvement, however, there are still several issues.

1. In Abstract, the main findings of this manuscript should be stated.

2. In Introduction, the description is a little odd. Why to give Analytical continuation of Schwarzschild metric, Black holes in astrophysics, et al firstly? The research background should be firstly introduced.

3. The equation should be numbered.

4. The empirical analysis and numerical computation related to ROEGENIAN ECONOMICS should be performed.

5. This version is still lack of literature review. 

Author Response

Answer Review 1

English language and style were verified again.

1. In Abstract, the main findings were stated.

2. In Introduction, the research background was moved on the first place adding clarifying text. Now the introduction provide sufficient background and include all relevant references used by authors. The description of the sources of inspiration is in alphabetical order, far from „little odd”.

3. The equations are not numbered because we have another style of presentation of results.

4. The empirical analysis and numerical computation related to ROEGENIAN ECONOMICS was performed by introducing two figures realized in Maple (Fig.1. Plot entropy colored by internal political stability and Fig. 2. Plot internal political stability colored by entropy).

5. There are other papers connected especially to the economics, but they do not reflect our aims (roegenian economics). The decision „this version is still lack of literature review” is biased.

Date 4 May 2019

Reviewer 2 Report

Dear authors, good job! I'm happy with the revision. Thanks, I recommend to publish the paper as is.

Author Response

AnswerReview2

Thank you very much for your opinion: „good job! I'm happy with the revision; thanks, I recommend to publish the paper as it is”.

Date

04 May 2019

Round 2

Reviewer 1 Report

The authors have made certain improvements according to the comments. However, there aer some issues, which the authors insist that they are not nassecary to be revised.

This paper has gone to several round reviews. Overall, the style of this manuscript is a liitle different from other papers. But the authors think it is this manuscript style. I donnot think so.

Form my view, this manuscript dose not still meet the publishing standard. However, the authors always make several revisions, not all the revisions related to my comment. I don't intend to review this manuscript again. Whether it is accepted or not, will be determined by other reviewers and editors.

This manuscript is a resubmission of an earlier submission. The following is a list of the peer review reports and author responses from that submission.

Round 1

Reviewer 1 Report

The paper is really interesting. It intersect thermodynamics with economics, with a novel vocabulary and novel derivations. I strongly recommend publication.

However, before formally accepting the paper, the authors should face some relevant criticisms.

1) Please, remove from the abstract the sentence claiming that the idea is totally new. Clearly, a scientific paper is new.

2) The authors should write a proper Introduction, where their contribution is clearly stated and reported in the respect of the existing literature.

3) The relationship between thermodynamic and economics should be described with more details. I don't underestand the economic meaning of the gravity constant, the light speed and the Plank constant. For example: in physics, nothing can be faster than the light speed. Does an analogous rule hold for the maximum universal exchange speed? This quantity cannot be given under the empirical experience, but as a universal statement. The authors should elaborate on this. Later, the authors assume c=1. Is it reaonable under an economic/physics perspective?

4) In general, the paper needs an effort to provide an economic sense to all the relations coming from the theoretical physics world.

5) Please, check the English of the paper. Some typos undermine the reading.

Author Response

Answer for Rev 1 – Entropy

1)    The abstract was improved mathematically and linguistically.

2)    A proper Introduction was realized underlying our contributions, reported to the existing literature.

3)    The relation between Thermodynamics and Economics was clarified. We added the following

Remark: In our idea the gravity constant and the Planck constant are related to the quantization of the dynamic quantities related to the microscopic world. This aspect find a scientific motivation also into social sciences as economics: in mechanics and quantum physics the same energies which act by producing effects and externalities in the real world at the same time playing a role in socio-economic mechanism designs in the real life side. The parallelism between the light speed and a concrete quantity as universal exchange speed, is rational and absolutely well- founded. This exchange-type could model the digital economic transitions among agents by world wide web. In our model we have fixed this constant to 1 (geometrized units), with the main aim to discuss about the scientific role played by this parameter which in this case does produce no efforts.

Remark: Since the total investment is a subunit percent of GDP, we accept $\frac{\mathcal{I}}{Y}<< 1$. Then $\left(1-\frac{\mathcal{I}^2}{Y^2}\right)^{1/2}$ is approximated by

$1-\frac{1}{2}\frac{\mathcal{I}^2}{Y^2}$, again by $1$, and hence the economic entropy is estimated by $E \cong 4\pi Y^2$. Based on UN and IMF estimations, the United States has the largest GDP in the world at

US\$ $18,624,475$ trillion (UN) and US\$ $21,410,230$ trillion (IMF). The second largest GDP is China's at \$ $11,218,281$ (UN) and \$ $15,543,710$ (IMF). Automatically, we have estimations for entropy

of particular economic Reissner-Nordstr\" om 3D Black Holes, as for example $E_{US}$ and $E_{C}$.

4)    The new version of the paper cover almost all economic senses of the relations coming from theoretical physics.

5)    Prof. Dr. Valeriu Prepelita (specialist in mathematics and English language) and Prof. Dr. Brandusa Prepelita-Raileanu (expert in English language) check the use of the English language in the paper. We've attached a pdf with linguistic proofing.

Thank you very much to both referees who have read the manuscript carefully and have made pertinent suggestions. As a result of the suggestions we completed the manuscript and removed the ambiguities.

Statement: In this paper we refer to Roegenian economics, based on our dictionary (morphism), enlarging the point of view of  Econophysics. The authors of this work are not just beginners, see our papers \cite{[20]}-\cite{[34]} and especially C. Udriste, M. Ferrara, D. Zugravescu, F. Munteanu, 2012, Controllability of a nonholonomic macroeconomic system, J. Optim. Theory Appl., 154, 3, pp. 1036-1054. Papers [33] and [34] clarify ideas for applicability in the real economy.

Reviewer 2 Report

This manuscript focuses on the Entropy of Reissner-Nordstrom 3D Black Hole in Roegenian Economics, which is an interesting topic. My comments are listed as bellows.

1. This manuscript is more like a report other than a paper, so the formulation and structure organization need to be improved.

2. The main contributions of this manuscript should be given.

3. ‘Dictionary’ has appeared many times in this manuscript, and what is the meaning of ‘Dictionary’?

4. The reference lumps, such as [20]-[34], [29]-[33] need to be avoided, especially in the first section.

5. Literature review should be conducted.

6. The empirical analysis on Roegenian economics should be conducted using the proposed theory in this manuscript.

7. Is there Entropy of Reissner-Nordstrom 3D Black Hole for other countries, such as US, China?

Author Response

Answer for Rev 2 – Entropy

1)    - 2) The manuscript was improved, regarding formulation and structure, underlining the main contributions. A proper Introduction was realized underlying our contributions, reported to the existing literature.

3)    New text: In our sense, the dictionary means a table that lists the words of a thermodynamic language and their correspondents in economics, associating "elements" that behave similarly. It is similar to a morphism (a structure-preserving map from one mathematical structure to another one of the same type). The information about usage of the dictionary is given in the text of the paper. Now “dictionary” does not appear many times.

4)    The reference lumps were eliminated.

5)    The literature was properly re-quoted.

6)    – 7) The relation between Thermodynamics and Economics was clarified. We added the following

Remark: In our idea the gravity constant and the Planck constant are related to the quantization of the dynamic quantities related to the microscopic world. This aspect find a scientific motivation also into social sciences as economics: in mechanics and quantum physics the same energies which act by producing effects and externalities in the real world at the same time playing a role in socio-economic mechanism designs in the real life side. The parallelism between the light speed and a concrete quantity as universal exchange speed, is rational and absolutely well- founded. This exchange-type could model the digital economic transitions among agents by world wide web. In our model we have fixed this constant to 1 (geometrized units), with the main aim to discuss about the scientific role played by this parameter which in this case does produce no efforts.

Remark: Since the total investment is a subunit percent of GDP, we accept $\frac{\mathcal{I}}{Y}<< 1$. Then $\left(1-\frac{\mathcal{I}^2}{Y^2}\right)^{1/2}$ is approximated by

$1-\frac{1}{2}\frac{\mathcal{I}^2}{Y^2}$, again by $1$, and hence the economic entropy is estimated by $E \cong 4\pi Y^2$. Based on UN and IMF estimations, the United States has the largest GDP in the world at

US\$ $18,624,475$ trillion (UN) and US\$ $21,410,230$ trillion (IMF). The second largest GDP is China's at \$ $11,218,281$ (UN) and \$ $15,543,710$ (IMF). Automatically, we have estimations for entropy

of particular economic Reissner-Nordstr\" om 3D Black Holes, as for example $E_{US}$ and $E_{C}$.

8)    Prof. Dr. Valeriu Prepelita (specialist in mathematics and English language) and Prof. Dr. Brandusa Prepelita-Raileanu (expert in English language) check the use of the English language in the paper. We've attached a pdf with linguistic proofing.

Thank you very much to both referees who have read the manuscript carefully and have made pertinent suggestions. As a result of the suggestions we completed the manuscript and removed the ambiguities.

Statement: In this paper we refer to Roegenian economics, based on our dictionary (morphism), enlarging the point of view of  Econophysics. The authors of this work are not just beginners, see our papers \cite{[20]}-\cite{[34]} and especially C. Udriste, M. Ferrara, D. Zugravescu, F. Munteanu, 2012, Controllability of a nonholonomic macroeconomic system, J. Optim. Theory Appl., 154, 3, pp. 1036-1054. Papers [33] and [34] clarify ideas for applicability in the real economy.

Round 2

Reviewer 1 Report

Dear Authors,

thanks a lot for your effort. Unfortunately, I think that the paper is not ready to be published.

Try to read your new Introduction, and tell me if you understand what is the scientific contribution of your study. Introduction is not clear. I need to have a complete view of what is the content of the paper only by reading it.

Thus, I recommend that the paper is accepted only after a proper presentation of the research in the Introductory section.

Author Response

Letter 2 for Rev 1

On your advice, we have improved the content of the paper, especially the introduction, and citations.

From the abstract, the sentence referring to our work was excluded.  To cover the need to have a complete view of the paper in introduction, we reformulated all things. Finally, a reformulation of the objectives of the paper was introduced.

Authors

Reviewer 2 Report

The paper has been improved. However, there are several works need to be done.

Introduction section is too short, whch should be enriched, and the main contribution should be discussed in details in this section.

Literature review has not been conducted.

In the revised paper, I have not seen the empirical analysis on Roegenian economics.

Author Response

Letter 2 for Rev 2

On your advice, we have improved the content of the paper, especially the introduction, and citations.

From the abstract, the sentence referring to our work was excluded.  To cover the need to have a complete view of the paper in introduction, we reformulated all things. Finally, a reformulation of the objectives of the paper was introduced.

The empirical analyses on Roegenian economics was specified at Economic cycles of Carnot type [34], and at Phase diagram for Roegenian economics [33].

In Section “Economic Reissner-Nordstrom (RN) 3D black hole” we gave formulas and their approximations for black hole entropy and internal political stability in the Roegenian economics, highlighting the situation in US and China. Processing information with “complexplot3d” clarifies some of the empirical analysis. All this was commented on in the new introduction (over 1 page).

Authors